# Association between Walking Pace and Diabetes: Findings from the Chilean National Health Survey 2016–2017

**DOI:** 10.3390/ijerph17155341

**Published:** 2020-07-24

**Authors:** Igor Cigarroa, María José Espinoza-Sanhueza, Nicole Lasserre-Laso, Ximena Diaz-Martinez, Alex Garrido-Mendez, Carlos Matus-Castillo, María Adela Martinez-Sanguinetti, Ana Maria Leiva, Fanny Petermann-Rocha, Solange Parra-Soto, Yeny Concha-Cisternas, Claudia Troncoso-Pantoja, Miquel Martorell, Natalia Ulloa, Heather Waddell, Carlos Celis-Morales

**Affiliations:** 1Escuela de Kinesiología, Facultad de Salud, Universidad Santo Tomás, Santiago 1015, Chile; yenyconchaci@santotomas.cl; 2Escuela de Enfermería, Facultad de Salud, Universidad Santo Tomás, Santiago 1015, Chile; mariaespinozasa@santotomas.cl; 3Escuela de Nutrición y Dietética, Facultad de Salud, Universidad Santo Tomás, Santiago 1015, Chile; nlasserre@santotomas.cl; 4Grupo de Investigación en Calidad de Vida, Departamento de Ciencias de la Educación, Facultad de Educación y Humanidades, Universidad del Biobío, Chillán 1180, Chile; xdiaz@ubiobio.cl; 5Departamento de Ciencias del Deporte y Acondicionamiento Físico, Universidad Católica de la Santísima Concepción, Concepción 2850, Chile; agarrido@ucsc.cl (A.G.-M.); cmatus@ucsc.cl (C.M.-C.); 6Instituto de Farmacia, Facultad de Ciencias, Universidad Austral de Chile, Valdivia 1954, Chile; mmartin3@uach.cl; 7Instituto de Anatomía, Histología y Patología, Facultad de Medicina, Universidad Austral de Chile, Valdivia 1954, Chile; aleiva@uach.cl; 8Institute of Health and Wellbeing, University of Glasgow, Glasgow G12 8QQ, UK; f.petermann-rocha.1@research.gla.ac.uk (F.P.-R.); 2489076P@student.gla.ac.uk (S.P.-S.); h.waddell@sms.ed.ac.uk (H.W.); Carlos.Celis@glasgow.ac.uk (C.C.-M.); 9British Heart Foundation Glasgow Cardiovascular Research Centre, Institute of Cardiovascular and Medical Sciences, University of Glasgow, Glasgow G12 8QQ, UK; 10Pedagogía en Educación Física, Facultad de Educación, Universidad Autónoma de Chile, Talca 2203, Chile; 11CIEDE-UCSC, Departamentos de Salud Pública, Facultad de Medicina, Universidad Católica de la Santísima, Concepción 2850, Chile; ctroncosop@ucsc.cl; 12Departamento de Nutrición y Dietética, Facultad de Farmacia, Universidad de Concepción, Concepción 1290, Chile; martorellpons@gmail.com; 13Centro de Vida Saludable, Universidad de Concepción, Concepción 1290, Chile; nulloa@udec.cl; 14Depto. de Bioquímica Clínica e Inmunología, Facultad de Farmacia, Universidad de Concepción, Concepción 1290, Chile; 15Centro de Investigación en Fisiología del Ejercicio (CIFE), Universidad Mayor, Santiago 2422, Chile; 16Laboratorio de Rendimiento Humano, Grupo de Estudio en Educación, Actividad Física y Salud (GEEAFyS), Universidad Católica del Maule, Talca 2203, Chile

**Keywords:** walking pace, diabetes mellitus, glucose, glycosylated haemoglobin A, health surveys, Chile (MeSH)

## Abstract

Background: Walking pace is a well-known indicator of physical capability, but it is also a strong predictor of type 2 diabetes (T2D). However, there is a lack of evidence on the association between walking pace and T2D, specifically, within developing countries such as Chile. Aim: To investigate the association between self-reported walking pace and T2D in the Chilean adult population. Methods: 5520 Chilean participants (aged 15 to 90 years, 52.1% women) from the Chilean National Health Survey 2016–2017 were included in this cross-sectional study. Both walking pace (slow, average, and brisk) and diabetes data were collected through self-reported methods. Fasting blood glucose (reported in mg/dl) and glycosylated haemoglobin A (HbA1c) scores were determined via blood exams. Results: In the unadjusted model, and compared to people who reported a slow walking pace, those with average and brisk walking pace had lower blood glucose levels (β = −7.74 mg/dL (95% CI: −11.08 to −4.40) and β = −11.05 mg/dL (95% CI: −14.36 to −7.75), respectively) and lower HbA1c (β = −0.34% (95% CI: −0.57 to −0.11) and β= −0.72% (95% CI: −0.94 to −0.49)), respectively. After adjusting for sociodemographic, Body Mass Index and lifestyle factors, the association between glycaemia and HbA1c remained only for brisk walkers. Both the average and brisk walker categories had lower odds of T2D (OR: 0.59 (95% CI: 0.41 to 0.84) and (OR 0.48 (95% CI: 0.30 to 0.79), respectively). Conclusion: Brisk walkers were associated with lower blood glucose and HbA1c levels. Moreover, average to brisk walking pace also showed a lower risk for T2D.

## 1. Introduction

Type 2 Diabetes (T2D) is a rising public health problem worldwide, specifically within the Chilean adult population. On a global scale, T2D affects 1 in 11 adults between the ages of 20 and 79 years old (currently 425 million adults worldwide live with diabetes) [1]. According to the International Diabetes Federation (IDF) the prevalence of T2D in Central and South America will increase to affect 62% of the population (from 26 to 42 million people) between 2017 and 2045 [2]. The Chilean National Health Survey 2016–2017 (CNHS 2016–2017) reported that, currently, 12.6% of the population has T2D [3]. This statistic is extremely concerning given that the World Health Organization (WHO) has identified that the direct health costs associated with T2D have reached a staggering US $827 billion per annum. Moreover, the indirect health costs are reported to be US $1.7 billion [4].

Although physical capability markers are generally used to predict morbidity and mortality in older adults [5,6], there is increasing evidence that walking pace could also be a strong predictor of health outcomes among middle-aged adults [7,8,9]. In a recent UK Biobank study, >500,000 participants were recruited between the ages of 40 and 70. This study reported that individuals with a brisk walking pace had a lower risk of developing cardiovascular (38%), respiratory diseases (42%), chronic pulmonary obstruction disease (74%) and premature mortality (21%) compared to slow walkers [7]. Moreover, walking pace is also a strong predictor of health in older adults. Recent international reports suggesting that walking pace is an important risk factor associated with fatigue, sarcopenia, osteoarthritis, disability, hospitalisation, falls, premature mortality, cancer, depression, cognitive impairment, high blood pressure, obesity, and T2D in older adults [10,11,12,13,14,15].

However, most of the current evidence about walking pace as a predictor of health outcomes has been generated within developed countries and does not extend to developing countries, such as Chile. In fact, only two studies have evaluated the association of walking speed with health outcomes in South America (one of them in Chile) [16,17]. Over the past two decades, Chile has undergone important health and epidemiological transitions. Currently, Chile is ranked third in Latin America for its prevalence rates of T2D, hypertension and obesity. Additionally, 24% of the population is reported to be physically inactive [18], increasing this prevalence with age [19]. Therefore, different strategies are needed to increase physical activity levels and maximise its various health benefits [20]. As walking pace could be a free, cheap, and feasible alternative to increase those health benefits, especially for those with T2D [20]. This study aimed to investigate the association of walking pace with T2D and diabetes biomarkers in a national representative sample of Chilean adults.

## 2. Materials and Methods

### 2.1. Study Design

This cross-sectional population study is composed of 6233 participants recruited from the Chilean National Health Survey conducted between 2016–2017 (CNHS 2016–2017). The CNHS 2016–2017 was conducted within Chilean households with participants over 15-years-old. Out of 6233 who took part in the CNHS only 5520 participants had blood glucose and glycosylated haemoglobin tests.

The present study was conducted according to the guidelines laid down in the Declaration of Helsinki, and the CNHS 2016–2017 has been reviewed by the Ministry of Health and ethically approved by the School of Medicine of the Pontifical Catholic University of Chile (16-019). All participants of the CNHS 2016–2017 provided written consent prior to participation [3].

### 2.2. Walking Pace

Walking pace was determined through the following questions “How would you describe your usual walking pace?” from which participants were asked to select one of the following three walking pace options slow, average, or brisk pace. The time allocated for physical activity (PA) related to transportation (walking, cycling) and moderate or vigorous intensity activities during leisure and/or work, were determined according to the GPAQ analysis guide (Global Physical Activity Questionnaire v2) [21,22]. To estimate total PA levels, the variables were expressed in METs (Metabolic equivalent of Task). An energy expenditure <600 MET/minutes/week, or its equivalent of 150 min of moderate to vigorous-intensity PA or 75 min of vigorous-intensity PA per week or its combination, was considered as a cut-off point for physical inactivity, according to the WHO recommendations and specifications of the GPAQ analysis guide [21,22]. Sedentary levels were determined by the self-report of time allocated to activities that involve sitting or reclining during free time or work (time sitting at the computer, watching television, travelling by bus, train, car, etc.) [21,22].

### 2.3. Diabetes Mellitus

The diagnosis of T2D was determined by the presence of any of the following three criteria: (a) Self-reported medical diagnosis of T2D, (b) Being under medical prescription for T2D, (c) Having fasting baseline glycaemic values >126 mg/dL [3].

### 2.4. Sociodemographics, Health, and Lifestyle

The sociodemographic variables age, sex (male or female), age group (15–37, 37–56, or >56 years), educational level (primary < 8, secondary 8 to 12 or higher education > 12 years), area of residence (rural or urban) and lifestyle variables: smoking habits, alcohol use (AUDIT scale), salt, fruit and vegetable intake, sleep hours and self-perception of health and personal wellbeing were obtained through the application of validated questionnaires of the CHNS 2016–2017 [3].

Weight was measured using a digital scale (Tanita HD-313^®^, Tokyo, Japan) and Height with a height rod in participant homes. Participants did not wear shoes and were dressed in light clothing for both measurements. These were both carried out through standardised methods and by trained nurses or midwives, as described elsewhere^3^. Nutritional status and body fat were determined according to Body Mass Index (BMI) based on the cut-off points of the WHO: underweight: <18.5 kg/m^2^; normal weight: 18.5–24.9 kg/m^2^; overweight: 25.0–29.9 kg/m^2^ and obesity: ≥30.0 kg/m^2^. Central obesity was defined as waist circumference (WC) ≥88 cm for women and ≥102 cm for men [23].

Blood glucose levels were determined according to HbA1c (glycosylated haemoglobin A) and glycaemia, through a blood sample, which was drawn after 11 h of fasting by trained nurses. The results were classified according to the parameters proposed by the American Diabetes Association (ADA): Normal HbA1c: 5.7%; HbA1c 5.7–6.4%: pre-diabetes; HbA1c greater than 6.5%: T2D [24].

### 2.5. Statistical Analyses

All analyses were performed with STATA 15 software (Statacorp; College Station, TX, USA). The analyses were performed by weighting the survey to the total national population, as suggested in the CNHS 2016–2017. Significant differences were accepted with a *p*-value < 0.05.

The characterisation data of the population studied were presented as weighted means for continuous variables and as a weighted prevalence for categorical variables with their corresponding 95% confidence intervals (95% CI).

The association of walking pace with fasting glucose and HbA1c were investigated using linear regression analyses. Data were presented as β-coefficient and its 95% CI. The association between walking pace and T2D was investigated with logistic regression analyses. Those who reported being “slow walkers” were the reference group, these results were reported as odds ratios (OR) and their 95% CI.

All analyses were incrementally adjusted according to different confounding factors. Model 0 was unadjusted; Model 1 was adjusted for sociodemographic factors such as age, sex, educational level and place of residence (urban/rural); Model 2 was additionally adjusted for BMI; finally Model 3 adjusted additionally adjusted for lifestyles behaviours including smoking, alcohol, sleeping behaviour, sitting time, and fruit and vegetable intake.

## 3. Results

Table 1 shows the characteristics of people by walking pace category. Approximately more than half of the participants reported an average walking pace (55.5%), whereas only 22.2% reported a slow walking pace. Compared to slow walkers, those who reported being average and brisk walkers were younger (41.0 and 39.1 years, respectively). Slow walking pace was more commonly reported in women than in men (60.8 versus 39.2%, respectively) and individuals with a lower education level. Individuals living in urban settings reported a higher prevalence of brisk walking compared to those living in rural settings. Slow walkers reported consuming more salt, less alcohol and more fruit and vegetables. They also had a lower prevalence of being regular smokers. Total PA, transport-related PA, and moderate-intensity physical activity were higher in brisk than slow walkers. However, vigorous-intensity PA was higher in slow walkers than brisk walkers. Additionally, slow walkers had a higher prevalence of obesity compared to brisk walkers (Table 1).

The associations of walking pace with fasting glucose and HbA1c are shown in Table 2. For the unadjusted model average and brisk walkers had lower levels of fasting glucose (β: −7.74 and −11.05 mg/dL, respectively) compare to slow walkers. Similar results were observed for HbA1c (β: −0.34 and −0.72% for average and brisk walkers compare to slow walkers). However, after adjusting the model, for confounding factors, only brisk walkers had lower levels of fasting glucose and HbA1c compared to slow walkers. (Table 2). Besides, a lower glycaemia and HbA1c trend values for each rise in walking speed category were observed (Figure 1).

The association between different walking pace and T2D is presented in Table 3. Compared to slow walkers, those who reported being average or brisk walkers had a lower risk of T2D (OR: 0.32 and 0.22, respectively). When these analyses were adjusted for major confounding factors, the associations were slightly attenuated but remained statistically significant (Table 3 and Figure 2).

## 4. Discussion

### 4.1. Walking Pace and Diabetes Biomarkers

Brisk walking pace was associated with lower blood glucose and HbA1c compared to those with a slow walking pace. Both average and brisk walkers had a lower risk of T2D in comparison to slow walkers. These associations were independent of age, sex, BMI, and other lifestyle behaviours.

### 4.2. What Was Already Known About This Subject?

Our findings are in agreement with international evidence, where average and brisk walking pace were associated with lower glucose levels compared to slow walkers [25,26]. However, for HbA1c, the evidence in the literature is discordant [25]. In this study, an association was observed between walking pace and HbA1c, mainly for brisk walkers. However, other studies have not found an association between walking pace and HbA1c, despite showing an association between walking pace and fasting glucose [27].

The associations found between walking pace and T2D agree with international existing evidence [25,26,27,28,29,30]. However, other studies have shown that walking pace is also a strong predictor of cardiovascular disease, cancer, and premature mortality in middle age [7,8,9,31] as well as cardiovascular risk, cardiovascular events, and risk of death in older adults [32,33,34]. In addition, walking pace has been used to estimate years of physical functioning lost in older adults and has been associated with both low socioeconomic status and risk factors for non-communicable diseases (among them T2D) [35].

Concerning Latin American literature, risk factors associated with T2D [36] and walking pace [37,38] have been studied as a functional marker in the population. However, the association between walking pace and health outcomes has been poorly studied [16,17]. In this context, our findings are consistent with a study carried out in Peru, in which a slow walking pace was associated with a higher prevalence of T2D. Nevertheless, the Peruvian study was conducted in an older population and, therefore, a fair comparison cannot be performed [16].

### 4.3. How Does This Study Contribute to Science?

If causal, the early assessment of walking pace could help as a cardiovascular and metabolic risk assessment in older adults [34]. More specifically, as a tool to identify individuals who have a high diabetic risk. Therefore, walking pace could be used as a screening question in primary care settings to identify a high-risk individual who would benefit from further metabolic screening but also physical activity intervention. Moreover, walking pace has been suggested as an easy-to-apply and low-cost screening tool, with important prediction ability for cardiovascular diseases, cancer, and premature mortality [39]. Besides, walking pace could be a simple, safe, free, cheap and feasible way to increase physical activity and maximise its health benefits in T2D population. Even more, if we consider that physical inactivity and higher levels of sitting time are two of the main modifiable risk factors associated with T2D [36] and that people with T2D have higher sedentary physical activity patterns and lower levels of physical activity than people without T2D [20].

### 4.4. Strengths and Limitations

This is the first study in Chile that determines the relationship between walking pace, diabetes biomarkers, and prevalence of T2D. In addition, fasting glucose and HbA1c have been measured using a standardised protocol, and prevalence of diabetes has been derived using WHO criteria. We were also able to adjust our analyses for a comprehensive list of confounding factors (BMI, sociodemographic factors, and lifestyle factors). However, we cannot remove the effect of confounding due to waist circumference and physical activity, as this might have resulted in multicollinearity issues. Other potential confounders, such as functional capacity, muscle strength, and balance, were unfortunately not collected in the CNHS. Finally, we cannot eliminate the potential effect of multimorbidity on our findings: people who reported a slow walking pace may also have other chronic diseases that may limit their walking pace. Although the walking pace was self-reported, previous studies have shown that this simple question is a significant predictor of health [32]. Another aspect to consider is that due to the cross-sectional nature of our study, our results cannot prove causality. However, previous evidence from randomised controlled trials suggests that walking is associated with better glycaemic control in healthy and diabetic patients [40,41,42].

## 5. Conclusions

Brisk walking pace was associated with a lower concentration of blood glucose and HbA1c compared to those with slow walking pace in the Chilean population. Besides, an average or brisk walking pace was associated with a lower likelihood of T2D compared to slow walkers.

## Figures and Tables

**Figure 1 ijerph-17-05341-f001:**
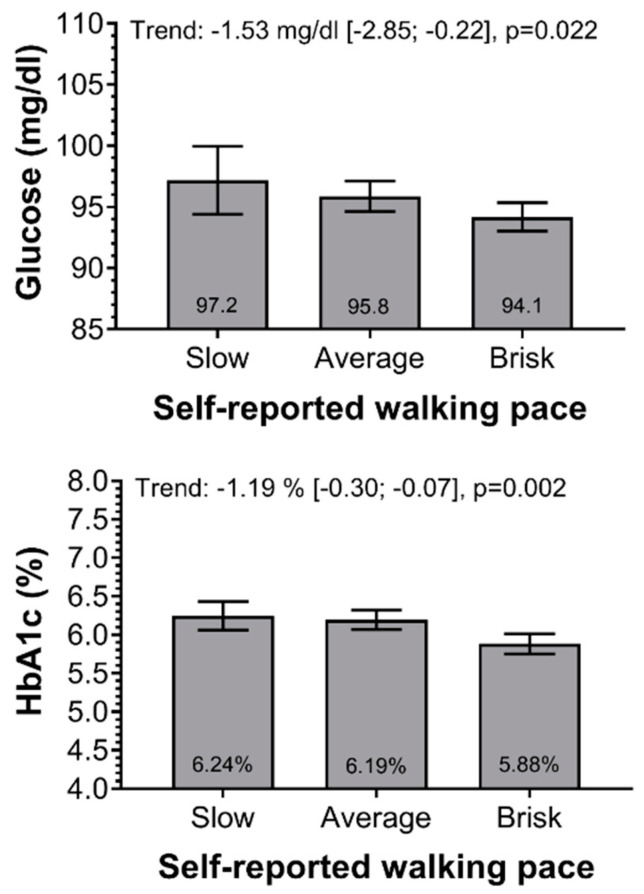
Association of walking pace with fasting glucose and HbA1c. Data presented as adjusted means and its 95% CI. Slow walking pace was considered as a reference value. Results were adjusted for model 3—age, sex, educational level and residence setting (urban/rural), BMI, smoking, alcohol, sleep duration, fruit and vegetable intake, and sitting time. The trends were estimated with regression analyses and presented as β-coefficient and 95% CI for the outcome per one category change in walking pace. HbA1c = Glycosylated Haemoglobin.

**Figure 2 ijerph-17-05341-f002:**
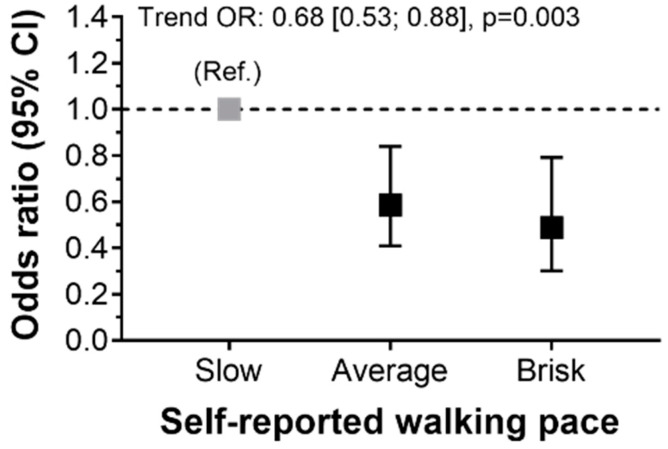
Association between walking pace and T2D. Data presented as odds ratio (OR) and its 95% confidence interval. Slow walking pace was considered as a reference value (Ref). Results were adjusted for model 3—age, sex, educational level and residence setting (urban/rural), BMI, smoking, alcohol, sleep duration, fruit and vegetable intake, and sitting time.

**Table 1 ijerph-17-05341-t001:** Characterisation of the population by walking pace.

Variables	Slow Pace	Average Pace	Brisk Pace
n (%)	1371 (22.2)	3429 (55.5)	1383 (22.4)
Sociodemographic			
Age (years) *	55.7 (53.6; 57.8)	41.0 (40.0; 42.0)	39.1 (37.7; 40.5)
Sex, %			
Women	60.8 (55.7; 65.7)	47.4 (44.2; 50.5)	51.1 (46.3; 55.9)
Men	39.2 (34.2; 44.2)	52.6 (49.4; 55.7)	48.9 (44.1; 53.7)
Place of residence, %			
Urban	84.7 (81.6; 87.3)	88.7 (87.1; 90.2)	92.5 (90.5; 94.1)
Rural	15.3 (12.6; 18.3)	11.3 (9.8; 12.8)	7.5 (5.8; 9.5)
Educational Level, %			
≤8 years	36.8 (32.4; 41.4)	13.0 (11.1; 15.0)	9.6 (7.3; 12.6)
9–12 years	48.0 (43.0; 53.1)	59.0 (55.8; 62.2)	55.2 (50.3; 59.9)
>12 years	15.2 (11.6; 19.6)	28.0 (50.3; 59.6)	35.2 (30.7; 40.1)
Lifestyle			
Smoking, %			
Regular smoker	18.7 (14.9; 23.2)	26.3 (23.5; 29.4)	24.1 (20.2; 28.5)
Occasional smoker	6.4 (4.1; 9.7)	9.3 (7.5; 11.5)	7.8 (5.6; 10.7)
Ex-smoker	27.3 (23.4; 31.6)	24.7 (21.9; 27.6)	26.5 (22.4; 31.0)
Non-smoking	47.6 (42.5; 52.6)	39.7 (36.7; 42.7)	41.6 (37.0; 46.3)
Alcohol use, %			
High consumption (AUDIT)	4.3 (1.8; 10.2)	5.0 (4.0; 7.3)	6.1 (3.7; 9.8)
F&V intake, %			
Eats less than 5 F&V	87.7 (84.2; 90.5)	86.2 (83.6; 88.5)	81.5 (77.4; 85.0)
Salt intake (g/day) *	9.5 (9.2; 9.8)	9.1 (8.9; 9.2)	8.9 (8.6; 9.2)
Sleep hours, %			
7–8 h	45.2 (40.4; 50.3)	54.2 (50.6; 57.3)	52.5 (47.8; 57.3)
≥9 h	25.9 (21.6; 30.6)	25.6 (23.0; 28.5)	23.4 (19.7; 27.6)
≤6 h	28.9 (24.4; 33.8)	20.2 (17.7; 23.0)	24.1 (20.2; 28.4)
Physical Activity			
Total PA (MET/min/day) *	771.1 (633.1; 909.1)	1200.0 (1100.5; 1301.2)	1412.9 (1232.4; 1591.5)
Transport PA (min/day) *	46.6 (32.8; 60.4)	72.5 (64.2; 80.7)	78.5 (65.8; 91.1)
Sedentary Time (min/day) *	214.4 (196.9; 231.8)	198.8 (187.6; 210.1)	208.0 (190.3; 225.6)
Moderate PA (min/day) *	192.5 (162.7; 222.2)	231.0 (211.1; 250.9)	235.1 (202.9; 267.3)
Vigorous PA (min/day) *	261.9 (211.9; 311.8)	202.1 (181.2; 223.0)	220.8 (186.1; 255.5)
Physical Inactivity, %	38.8 (34.1; 43.6)	23.9 (21.3; 26.7)	18.9 (15.5; 22.8)
Adiposity			
Body weight (kg) *	76.3 (74.6; 78.1)	75.7 (74.7; 76.7)	74.7 (73.3; 76.2)
BMI (kg/m^2^) *	30.2 (29.6; 30.9)	28.4 (28.1; 28.7)	27.7 (27.3; 28.1)
Nutritional status, %			
Underweight	1.1 (0.0; 2.6)	1.5 (0.1; 2.6)	0.8 (0.0; 0.2)
Normal	21.8 (17.8; 26.4)	24.2 (21.6; 27.0)	27.0 (23.0; 31.4)
Overweight	29.9 (25.6; 34.6)	41.7 (38.6; 44.9)	42.4 (37.7; 47.3)
Obese	47.2 (42.1; 52.3)	32.6 (29.7; 35.6)	29.8 (25.6; 34.3)
Waist circumference (cm) *	98.3 (96.9; 99.8)	92.6 (91.7; 93.5)	91,3 (90.1; 92.4)
Central obesity, %			
>102 cm men and >88 cm women	59.0 (53.9; 63.9)	41.3 (38.3; 44.4)	39.2 (34.6; 43.9)

Data presented by average walking pace (slow, average, brisk) and 95% CI for continuous variables (*) and in % and 95% CI for categorical variables. PA = Physical Activity; BMI = Body Mass Index; F&V = Fruits and Vegetable; AUDIT = Alcohol Use Disorders Identification Test.

**Table 2 ijerph-17-05341-t002:** Association of walking pace with fasting glucose and glycosylated haemoglobin A (HbA1c).

Variables	Slow Pace	Average Pace		Brisk Pace	
		β (95% CI)	*p*-Value	β (95% CI)	*p*-Value
Fasting Glucose (mg/dL)					
Model 0	1.00 (Ref.)	−7.74 (−11.08; −4.40)	<0.0001	−11.05 (−14.36; −7.75)	<0.0001
Model 1	1.00 (Ref.)	−2.55 (−5.67; 0.56)	0.109	−4.84 (−7.91; −1.79)	0.002
Model 2	1.00 (Ref.)	−1.70 (−4.86; 1.34)	0.266	−3.64 (−6.72; −0.57)	0.020
Model 3	1.00 (Ref.)	−1.31 (−4.43; 1.80)	0.410	−3.00 (−5.99; −0.11)	0.049
HbA1c (%)					
Model 0	1.00 (Ref.)	−0.34 (−0.57; −0.11)	0.004	−0.72 (−0.94; −0.49)	<0.0001
Model 1	1.00 (Ref.)	−0.08 (−0.30; 0.13)	0.464	−0.40 (−0.61; −0.17)	<0.0001
Model 2	1.00 (Ref.)	−0.08 (−0.29; −0.14)	0.493	−0.39 (−0.62; −0.16)	0.001
Model 3	1.00 (Ref.)	−0.05 (−0.28; 0.18)	0.665	−0.37 (−0.60; −0.13)	0.002

Data presented as β-coefficient and its 95% CI by walking pace category estimated by linear regression analysis. Slow walking pace was considered as a reference value (Ref). Statistical analyses were incrementally adjusted: Model 0—unadjusted; Model 1—adjusted by sociodemographic factors (age, sex, educational level and residence setting (urban/rural); Model 2—was additionally adjusted for BMI; Model 3—was additionally adjusted for lifestyle factors (smoking, alcohol, sleep duration, fruit and vegetable intake, and sitting time). HbA1c = Glycosylated Haemoglobin.

**Table 3 ijerph-17-05341-t003:** Association between walking pace and Type 2 Diabetes (T2D).

Variable	Slow Pace	Average Pace		Brisk Pace	
		OR (95% CI)	*p*-Value	OR (95% CI)	*p*-Value
T2D					
Model 0	1.00 (Ref.)	0.32 (0.4; 0.43)	<0.0001	0.22 (0.15; 0.34)	<0.0001
Model 1	1.00 (Ref.)	0.54 (0.38; 0.77)	0.001	0.42 (0.26; 0.66)	<0.0001
Model 2	1.00 (Ref.)	0.58 (0.41; 0.83)	0.003	0.48 (0.31; 0.77)	0.002
Model 3	1.00 (Ref.)	0.59 (0.41; 0.84)	0.004	0.48 (0.30; 0.79)	0.004

Data presented as odds ratio (OR) and its 95% CI by walking pace category. Slow walking pace was considered as a reference value (Ref.). The OR and its respective 95% CI were determined by logistic regression. Statistical analyses were incrementally adjusted: Model 0—unadjusted; Model 1—adjusted by sociodemographic factors (age, sex, educational level and residence setting (urban/rural); Model 2—was additionally adjusted for BMI; Model 3—was additionally adjusted for lifestyle factors (smoking, alcohol, sleep duration, fruit and vegetable intake, and sitting time). T2D = Type 2 Diabetes.

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
