# Peer review of "Association between Walking Pace and Diabetes: Findings from the Chilean National Health Survey 2016–2017"

_ijerph, 2020, doi:10.3390/ijerph17155341_

Round 1

Reviewer 1 Report

After careful review of the study by Cigarroa et al, my recommendation for publication would be rejection. I would consider this manuscript to be in the bottom 50% of those I review. 

During my review, I noted many serious concerns, including

1) Complete lack of introduction and discussion to find the novelty. 

2) Minimal data as most of the physiological parameters to diagnosis the diabetes have been established.

3) Endnote problem. Please correct.

These notable issues make interpretation and meaningfulness of findings very challenging.

Author Response

Point 1: Complete lack of introduction and discussion to find the novelty.

Response: Thank you for sharing your feedback. We have modify the introduction and discussion to bring the novel aspect of the study to light such as lack of evidence in Chile, as well as in the rest of Latin American countries

Point 2:  Minimal data as most of the physiological parameters to diagnosis the diabetes have been established.

Response: Thank your comment. The study was carried out with the data provided by the Chilean National Health Survey (CNHS 2016-2017). In this survey, the data presented in the manuscript, are the parameters related to diagnosis the diabetes.

Point 3: Endnote problem. Please correct.

Response: Thank you for this suggestion. This has been amended.

Reviewer 2 Report

The study describing an association between walking pace in Chile can be considered T2D is relevant since it expands information on this risk factor from different types of populations.

However, it needs several items that need to be improved to increase the quality of the manuscript and the work. Below I list some of the main changes that must be made to the manuscript.

Introduction

Line 58 - Please move the reference number 2 to the end of the next sentence

Lines 62, 63 and others in the manuscript - Sometimes the authors write "diabetes mellitus" by T2D and others write full words. After the first presentation authors must be just use T2D in all text.

Line 70 and others in the manuscript - Please put a space between the word and the parentheses. Review the entire manuscript to normalize this action.

The space between lines is different in the Introduction and at the beginning of the method section.

Methods and Results

From the observation of Table 1, it is clear that the authors chose to present some results by absolute and relative frequencies of each variable in the different  walking pace categories. In this context, I think it would be important to conduct a univariate analysis (especially chi-square) that would make it possible to observe the differences between each of the walking pace groups. Multivariate analysis by logistic regression would help to observe potential intervening factors, mainly gender and age.  In this case I also suggest that the authors add a new column in the table showing the p value of each variable.

I believe the authors have switched the units of glucose values. They have values that seem to me to be in mmol / Liter and not mg / dL. This review is of fundamental importance to be done including the abstract!

Discussion

  In Table 1, the authors made no reference to other non-communicable chronic diseases that could have some level of influence on the results obtained. This type of information is relevant mainly considering that the sample included individuals from 18 to 90 years old and it is known that from 60 years of age, in developing countries like Chile many individuals have plurimorbidities. If it is not possible to include this information in the analyzes, please add a paragraph in the Discussion talking about this important limitation.  This pressumption is based in previous studies such as performed by  Celis-Morales et al.  Med Sci Sports Exerc. 2019 Mar;51(3):472-480, that showed, for example association between walking pace and prostate cancer.   English editing is mandatory!  

Author Response

Introduction

Point 1: Line 58 - Please move the reference number 2 to the end of the next sentence

Response: Thank you for this suggestion. This issue has been amended.

Point 2: Lines 62, 63 and others in the manuscript - Sometimes the authors write "diabetes mellitus" by T2D and others write full words. After the first presentation authors must be just use T2D in all text.

Response: Thank you. This issues has been amended troughout the manuscript.

Point 3: Line 70 and others in the manuscript - Please put a space between the word and the parentheses. Review the entire manuscript to normalize this action.

Response: A space between the words has been added.

Point 4: The space between lines is different in the Introduction and at the beginning of the method section.

Response: Thank you for bringing this to our attention. Both the introduction and the beginning of the methods sections now have the same space.

Methods and Results

Point 1: From the observation of Table 1, it is clear that the authors chose to present some results by absolute and relative frequencies of each variable in the different walking pace categories. In this context, I think it would be important to conduct a univariate analysis (especially chi-square) that would make it possible to observe the differences between each of the walking pace groups. Multivariate analysis by logistic regression would help to observe potential intervening factors, mainly gender and age.  In this case I also suggest that the authors add a new column in the table showing the p value of each variable.

Response: Thank you for this observation. However, in order to follow the new STROBE statement for cross-sectional studies, we did not perform any formal comparison for variables presented in Table 1. Considering that Table 1 aims is only to characterize the population by walking pace categories. Estimating any difference between subgroup could generate a type I error.

Point 2: I believe the authors have switched the units of glucose values. They have values that seem to me to be in mmol / Liter and not mg / dL. This review is of fundamental importance to be done including the abstract!

Response: Thank you for bringing this to our attention. We have check the glucose data and we confirm that the values are presented in mg/dl units. 

Response: Thank you for bringing this to our attention. We have check the glucose data and we confirm that the values are presented in mg/dl units. in addition, it was included in the abstract.

Discussion

Point 1: In Table 1, the authors made no reference to other non-communicable chronic diseases that could have some level of influence on the results obtained. This type of information is relevant mainly considering that the sample included individuals from 18 to 90 years old and it is known that from 60 years of age, in developing countries like Chile many individuals have plurimorbidities. If it is not possible to include this information in the analyzes, please add a paragraph in the Discussion talking about this important limitation.  This pressumption is based in previous studies such as performed by Celis-Morales et al.  Med Sci Sports Exerc. 2019 Mar;51(3):472-480, that showed, for example association between walking pace and prostate cancer.  

Response:  Thanks, we have amended the discussion section to highlight this limitation.

Point 2: English editing is mandatory!   

Response: We thanks for this observation. A native English speaker has read and proof-read the document.

Reviewer 3 Report

This is a very interesting and well written article. However, I have some minor remarks.

The introduction and most parts of the methodology are very clear. The fact that Chile has such a high prevalence of diabetes is alarming.

  1. I may assume that the researchers analyzed a data set and were not involve in data collection. However, there is a need for some information about the sampling methods, the procedures of measuring height and weight and blood testing.
  2. Please note that on line 89 it is written: “participants between the ages of 15 and [insert number].”
  3. It is written, in the beginning of the results that: “Table 1 shows the characteristics of people by walking pace category. Approximately more than 144 half of the participants reported an average walking pace (54.7%) whereas only 17.9% reported a slow 145 walking pace.” – the data in the table are similar to what is mentioned but not exactly the same.
  4. Table 4 presents results of a logistic regression, but the footnote mention by mistake “The beta coefficient and its respective 95% CI were determined by linear regression.”
  5. Under the heading of 4.3. How does this study contribute to science? The authors suggest that “The early assessment of walking pace could help identify participants with an adverse metabolic profile and more specifically individuals who have a high diabetic risk.” This is a too ambitious suggestion, as the study was a cross-sectional one, unless they present previous longitudinal studies to prove causality.

Author Response

Point 1: I may assume that the researchers analyzed a data set and were not involve in data collection. However, there is a need for some information about the sampling methods, the procedures of measuring height and weight and blood testing.

Response: Thank you for this suggestion. More information has been added in the methods section.

Point 1: Please note that on line 89 it is written: “participants between the ages of 15 and [insert number].”

Response: Thank you for bringing this to our attention. The number has been inserted.

Point 2: It is written, in the beginning of the results that: “Table 1 shows the characteristics of people by walking pace category. Approximately more than 144 half of the participants reported an average walking pace (54.7%) whereas only 17.9% reported a slow 145 walking pace.” – the data in the table are similar to what is mentioned but not exactly the same.

Response: Thank you for bringing this to our attention. This information was amended.

Point 3: Table 4 presents results of a logistic regression, but the footnote mention by mistake “The beta coefficient and its respective 95% CI were determined by linear regression.”

Response: Thank you for bringing this to our attention. This information was amended.

Point 4: Under the heading of 4.3. How does this study contribute to science? The authors suggest that “The early assessment of walking pace could help identify participants with an adverse metabolic profile and more specifically individuals who have a high diabetic risk.” This is a too ambitious suggestion, as the study was a cross-sectional one, unless they present previous longitudinal studies to prove causality.

Response: Thank you for sharing your thoughts. Although prospective evidence is better than cross-sectional evidence none of these study designs can prove causality. We have amended this statement, as follow “If causal, the early assessment of walking pace could help identify participants with an adverse metabolic profile and more specifically individuals who have a high diabetic risk.” In addition, references from randomized clinical trials were added suggesting that walking is associated with better glycemic control in healthy and diabetic patients.

Round 2

Reviewer 1 Report

The authors have been responsive to comments and the manuscript has not been greatly improved.

Actually, I have not figured out the characteristics of participants for diagnosis of diabetes stages and biochemical parameters including insulin, TC, TG, HDL-C, LDL-C, and HOMA-IR. 

Author Response

The authors have been responsive to comments and the manuscript has not been greatly improved.

Point: Actually, I have not figured out the characteristics of participants for diagnosis of diabetes stages and biochemical parameters including insulin, TC, TG, HDL-C, LDL-C, and HOMA-IR.

Response:

Thank you for this suggestion. As these biomarkers were out of the scope of the current study, we decided not to include it. However, the introduction and discussion have been updated and new references added to emphasize how novel the results are for South America.

Reviewer 2 Report

I consider that manuscript was improved and is able to publication

Author Response

Thank you very much for the trust placed in the manuscript.